# Accuracy of long-term volunteer water monitoring data: A multiscale analysis from a statewide citizen science program

**Kelly Hibbeler Albus**[1☯], **Ruthanne Thompson**[1☯]*, **Forrest Mitchell**[2‡], **James Kennedy**[1‡], **Alexandra G. Ponette-González**[3‡]

**1** Department of Biology, University of North Texas, Denton, Texas, United States of America, **2** Department of Entomology, Texas A&M University, College Station, Texas, United States of America, **3** Department of Geography and the Environment, University of North Texas, Denton, Texas, United States of America

☯ These authors contributed equally to this work.
‡ These authors also contributed equally to this work.
* ruthanne.thompson@unt.edu

## Abstract

An increasing number of citizen science water monitoring programs is continuously collecting water quality data on streams throughout the United States. Operating under quality assurance protocols, this type of monitoring data can be extremely valuable for scientists and professional agencies, but in some cases has been of limited use due to concerns about the accuracy of data collected by volunteers. Although a growing body of studies attempts to address accuracy concerns by comparing volunteer data to professional data, rarely has this been conducted with large-scale datasets generated by citizen scientists. This study assesses the relative accuracy of volunteer water quality data collected by the Texas Stream Team (TST) citizen science program from 1992–2016 across the State of Texas by comparing it to professional data from corresponding stations during the same time period. Use of existing data meant that sampling times and protocols were not controlled for, thus professional and volunteer comparisons were refined to samples collected at stations within 60 meters of one another and during the same year. Results from the statewide TST dataset include 82 separate station/year ANOVAs and demonstrate that large-scale, existing volunteer and professional data with unpaired samples can show agreement of ~80% for all analyzed parameters (DO = 77%, pH = 79%, conductivity = 85%). In addition, to assess whether limiting variation within the source datasets increased the level of agreement between volunteers and professionals, data were analyzed at a local scale. Data from a single partner city, with increased controls on sampling times and locations and correction of a systematic bias in DO, confirmed this by showing an even greater agreement of 91% overall from 2009–2017 (DO = 91%, pH = 83%, conductivity = 100%). An experimental sampling dataset was analyzed and yielded similar results, indicating that existing datasets can be as accurate as experimental datasets designed with researcher supervision. Our findings underscore the reliability of large-scale citizen science monitoring datasets already in existence, and their potential value to scientific research and water management programs.

**Data Availability Statement:** All relevant data are available from ODF: https://doi.org/10.17605/OSF.IO/52UJQ.

**Funding:** The author(s) received no specific funding for this work.

## Introduction

In an effort to address extensive surface water quality monitoring needs, many U.S. state and federal agencies utilize citizen science or volunteer water monitoring programs to collect monitoring data and to educate and involve the public in the management of their own watersheds [1, 2, 3, 4]. Citizen science is a term used to describe intentional participation of volunteers in the scientific process, and citizen science programs have seen rapid growth in the last decade [5, 6, 7]. Volunteer water quality monitoring programs are one of the most prevalent examples of citizen science, increasing in number since the 1970s. According to the National Water Quality Monitoring Council, there are currently over 1,720 groups across the U.S. conducting volunteer water monitoring and associated activities. Many of these programs are supported by regional regulatory agencies that often provide training, equipment, and quality assurance protocols [8, 9, 10, 11].

The benefits of involving citizen science in environmental research extend beyond data collection, with program involvement leading to increased public engagement in environmental stewardship, increased scientific literacy, and development of community-centered goals and public policy, for example [12, 13, 14]. With environmental monitoring efforts, however, the accuracy and applicability of the data collected are also a large component of a program's success, especially over the long-term [15, 16, 17, 18, 19]. Long-term water monitoring datasets have been shown to be especially useful for providing baseline information on streams, identifying target areas for further sampling, and providing evidence for land-use change impacts [1, 20, 21, 22].

One example of a long-term citizen science water monitoring program is the Texas Stream Team (TST), a statewide program with a network of trained volunteers collecting stream data across Texas since 1991. The program was initiated by the Texas Commission for Environmental Quality (TCEQ), the state agency responsible for statewide water quality management, to supplement professional monitoring efforts across the state and to increase public outreach and education about stream health. By their 25[th] anniversary in 2016, TST had involved over 8,600 citizen scientists actively monitoring 280 stations statewide, covering over 30500 km of Texas streams [23].

The TST is an example of a state-supported program in which a centralized staff coordinates with independent partner agencies. Partner agencies typically manage the volunteer monitoring efforts in their region, and host sampling kits and training events. TST volunteers complete a three-level training course, approved by TCEQ and utilizing U.S. Environmental Protection Agency (USEPA)-approved protocols, to receive their certification. TST maintains a Quality Assurance Project Plan (QAPP) that is reviewed regularly to adhere to current national standards for data collection and standardization [24]. The Texas program with professional oversight and quality assurance protocol is similar to those in other states, such as Colorado River Watch Network, Missouri Stream Team, Georgia Adopt-a-Stream, the Alabama Water Watch, and IOWATER [1, 21, 25, 11]. According to a recent survey by Stepenuck and Genskow [26] there are 345 such volunteer monitoring programs in 47 states, with each state having between one and 30 volunteer programs nested within. For a growing number of these programs, state regulatory agencies have begun to incorporate long-term volunteer-collected data into their official watershed reporting databases, but for many other programs there are no established uses for volunteer data.

Volunteer-collected data are not used in an official capacity in the State of Texas. Published literature reports that, despite quality assurance protocols, data collected by volunteer water monitoring programs are often underutilized by professionals and scientists due to concerns about the accuracy of data collected [27, 1, 20, 10, 11]. A growing number of studies have

compared data collected by volunteers to data collected by employed professionals as a proxy for determining accuracy and overall quality of volunteer data, but many of these studies are inconclusive, and do not include existing data collected by programs prior to the study [28, 29].

The goal of this research was to assess the relative accuracy of a long-term (1992–2016) volunteer (TST) water quality dataset by comparing it to professional data collected at statewide and local (i.e., city) scales in the State of Texas. We hypothesized that with increasing control over sampling times and a reduction in the spatial coverage of data collection, agreement between volunteer and professional data would increase. To address this hypothesis, we asked the following two questions using existing monitoring data: (1) Do statewide volunteer-collected water quality data differ significantly from data collected by professionals at corresponding sampling stations over the same time period? (2) Do volunteer-collected water quality data collected in the City of Denton from 2009–2017 differ significantly from data collected by professionals at corresponding sampling stations when both have sampled at that station within a 5-day period? In addition, we conducted an experiment in which samples collected by volunteers and professionals were paired in time and space. Using this experimental data, we asked: (3) Do volunteer-collected water quality data collected in the City of Denton from 2017–18 differ significantly from data collected by professionals at corresponding sampling stations when both have sampled at that station within a 5-day period, when the data has been collected as part of an experimental study?

In addition to being the first scientific study to utilize TST's long-term, statewide dataset, this study is also, to the best of our knowledge, the first in the literature to conduct a multi-scale comparison analysis of variance between volunteers and professionals. Further, we utilized existing datasets that cover multiple decades at both at statewide and local scales, as well as experimental data collected for this study. The results from this study can provide a more comprehensive understanding of the variations between volunteer and professional data over a volunteer monitoring program's entire history.

## Materials and methods

We selected the TST program to examine the accuracy of long-term, large-scale citizen science data collected under quality assurance plans given its extensive record of data collection and area of coverage, as well as concurrent sampling with professional agencies. TST is coordinated by a central staff but much of TST data collection relies on local organizations, such as river authorities and city municipalities to act as partners [24]. Partners determine sampling needs, develop a localized water monitoring plan, purchase water sampling kits that they maintain and check out to trained volunteers, communicate with and recruit volunteers, and host training sessions for new volunteers to become certified. To become a certified TST Citizen Water Quality Monitor, volunteers must complete a three-phase training course using a test kit that measures the physical and chemical parameters of water, with protocols that are aligned with a TCEQ-approved QAPP. TST-certified volunteers collect data under an approved monitoring plan. Data are verified and reported with the required associated metadata to the TST staff (QAPP) [24]. Participation in the program includes a commitment to monitor at least one location every month, at approximately the same time of day for one year, and to attend two quality-control sessions in the first year and one session per year thereafter. As such, much of the data from TST volunteers have been collected monthly. This nested program structure allows for large-scale data collection with dispersed resources but a quality assurance plan to maintain standard protocols [26].

Professional monitoring data are collected by a staff member or researcher employed by the TCEQ, or one of its partner agencies. TCEQ is responsible for all official surface water quality

monitoring in Texas and reporting standards to the EPA. TCEQ manages this through regional agencies certified to assist with efforts. These agencies conduct the majority of the statewide monitoring on behalf of TCEQ, and as per state regulations, the monitoring data they collect are made publicly accessible through their online database [30]. They are also certified under an EPA-aligned QAPP through TCEQ and, as such, they are provided with guidelines for sampling collection, equipment and data reporting. Although most of the monitoring data from the TCEQ database are collected on a quarterly basis, many of the stations have been monitored more frequently.

Throughout the 25+ years that TST volunteers have sampled stream sites in Texas, there are many instances when professional entities also sampled at the same stream locations, in some cases for decades at a time. TST and TCEQ sampling protocols are both under a QAPP in accordance with USEPA national standards and are assumed to be of consistent quality despite differences in specific sampling equipment or personnel, providing an opportunity for a statewide comparison analysis between volunteers and professionals.

Localized data from the City of Denton were evaluated to assess if the results from a single partner agency differ significantly from statewide results. The City of Denton is one of TST's partner agencies that operates under a QAPP and reports their findings to the state agency in an official capacity. The city is part of multiple watershed restoration efforts that require additional monitoring and has been utilizing TST volunteers to monitor many of the same sites as professional staff for over 10 years.

## Data collection and analysis

To evaluate the relative accuracy of existing volunteer data at statewide (1992–2016) and local (2009–2017) scales, we compared TST volunteer-collected water quality data with TCEQ/CRP professionally collected water quality data. The CRP Data Tool: https://www80.tceq.texas.gov/SwqmisWeb/public/crpweb.faces is publicly available. The queries for the dataset used the same dates and parameters for each query. For example, Start date: 01/01/1992; End Date 09/01/2016; Parameter Group: Field Water Quality; Basin 5-Sabine River. This was done for each of the 12 basins that were part of the study. This data was extracted to a Result data .txt file, which was then saved to an Excel data file.

We then compared three water quality parameters: dissolved oxygen (DO), pH and conductivity as these parameters are the most commonly assessed by both volunteers and professionals and for which there was sufficient sampling data for comparative analysis. Data were obtained directly from TST and City of Denton staff (2017) for this study. All TST volunteer data and TCEQ/CRP professional data are made publicly-available through state regulations and can be obtained through their online data portals. The professional data is available through the CRP Data Tool [31] and the TST volunteer data is available through their dataviewer [32].

Data were refined (cleaned, sorted by station and year, visualized) using Excel and ArcGIS (Fig 1). Samples were selected for analyses when samples were collected: 1) at a station for which both TST and TCEQ had a corresponding station (i.e., one with the same Station ID or located within 60 meters of the same segment), and 2) during a year in which both TST and TCEQ sampled stream segments two or more times (station year). Due to unknown sampling techniques, variable sampling dates and times, and the large geographic area covered, these data were not analyzed as paired samples for the statewide dataset. Instead, the data were pooled and analyzed by year, after determining that data were normally distributed for that year. At the local scale, the authors included only samples in which volunteers and professionals collected samples within five days of each other, and at identical sites. These paired samples

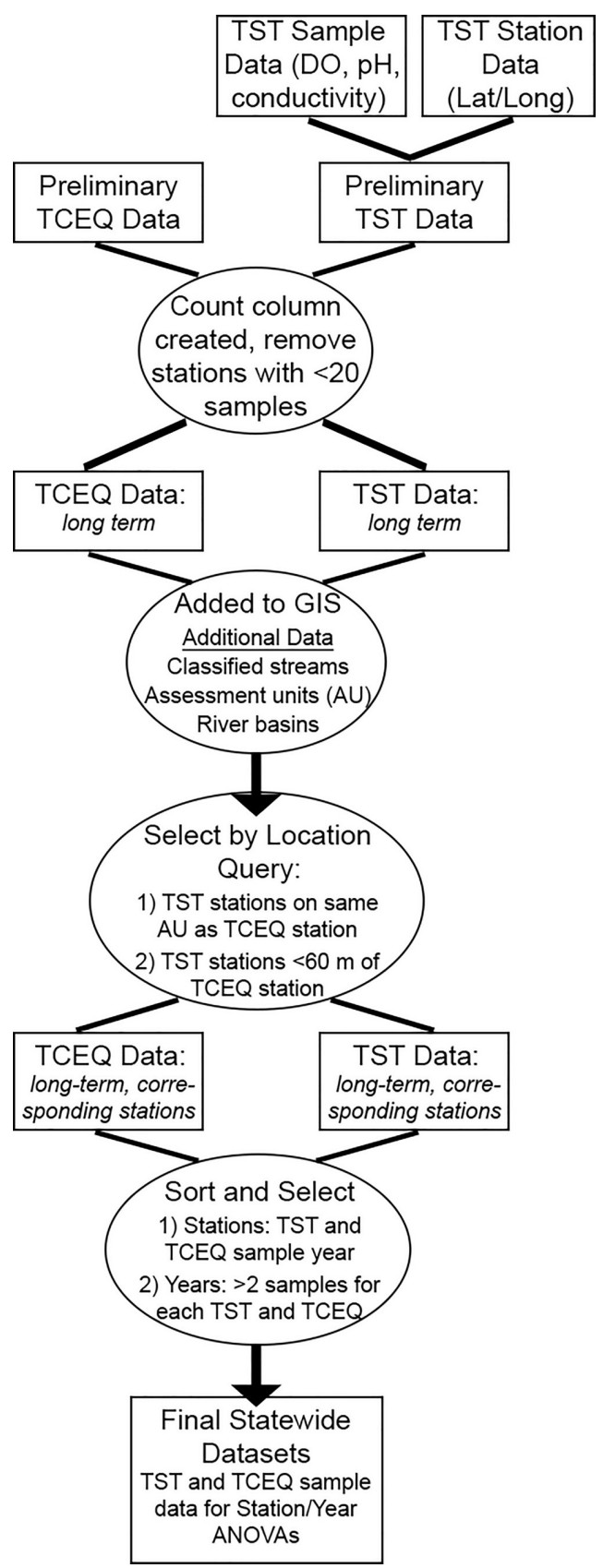

**Fig 1. Flowchart of data collection methods, statewide (1992–2016) datasets.** Step-by-step process to render large-scale, long-term volunteer (TST) and professional (TCEQ) datasets more comparable for analysis.

and localized geographic range allowed for less variation within the source datasets than at the state scale, allowing us to assess if increased control over sampling protocols increased the agreement between volunteers and professionals.

All data analyses were performed using SAS 9.4 (Statistical Analysis Systems, Cary, NC). An analysis of variance (ANOVA) was conducted to determine variance between the two source datasets at each station, for each station year, and for each of the three parameters. A univariate Kolmogorov-Smirnov D (KSD) test was also performed for each of the models to assess normal distribution of the residuals through homogeneity of variance. These were evaluated individually, and if not normally distributed, that station/year was removed from the final results. For local scale data, samples that did not have a normal distribution were log transformed. Local-scale pH data could not be log-transformed; thus, all pH analyses were conducted using a non-parametric Kolmogorov-Smirnoff Two-Sample (KST) test using the NPAR1WAY procedure. From the resulting ANOVAs, we summarized overall percent agreement between volunteer and professional data across all sites and years, for the statewide total, and for each of the three parameters, DO, pH and conductivity. Results by river basin and by years were also examined to assess differences by location and over time.

At the local scale data, an additional group ANOVA was run on the entire dataset to compare DO, pH and conductivity of TST volunteers to that of City of Denton professionals across all stations and years. Due to greater control over geographic range and sample times, the two source datasets were more directly comparable, and a group analysis was viable. This analysis was also conducted so that the results from the larger, existing dataset from City of Denton could be compared to the smaller experimental City of Denton dataset discussed in the following section. In this way, one local scale analysis compares directly to that of the largest, and most variable statewide dataset, and one to that of the smallest experimental dataset, allowing for a greater understanding of variation across scales.

We also collected experimental data, with a view to understand if data collected through a more controlled, experimental sampling design similar to those used in other studies [33], would generate results that were different from those of existing datasets. The experimental portion of this analysis was designed such that the sampling protocol was similar to that of the existing datasets in every regard except for professional sampling time. The volunteers, who were uninformed about the study, followed the regular TST protocol, continuing to sample according to their established monthly schedule and sample locations. These volunteers submitted their data to the City of Denton staff following standard procedure. The City of Denton staff were informed of the study and therefore timed their regular sampling to be within 5 days (and often within the same day) as the volunteers. Other than this change in timing, the City of Denton staff also continued to follow their standard sampling protocol.

These samples were collected from November 2017 through May 2018. In order for a sample to be included in this third analysis, the sample had to 1) be collected at a station for which both TST and City of Denton had a corresponding station (one with the same Station ID or same station location), and 2) the volunteer and professional samples had to have been collected within 5 days of each other, resulting in a dataset with paired samples. A group ANOVA was conducted to assess the variance between TST volunteers and City of Denton professionals across all stations and dates.

**DO–systematic bias adjustment.**   Upon review of the City of Denton volunteer and professional sampling data (both existing and experimental), we detected a difference between the volunteer and professional data, with the volunteer data consistently lower than the professional data at every station, and across all years). This reproducibility across all samples within the dataset indicated systematic bias or error, with a consistent magnitude, rather than a reflection of the actual variation in the dataset [34]. This was confirmed when visual assessment of

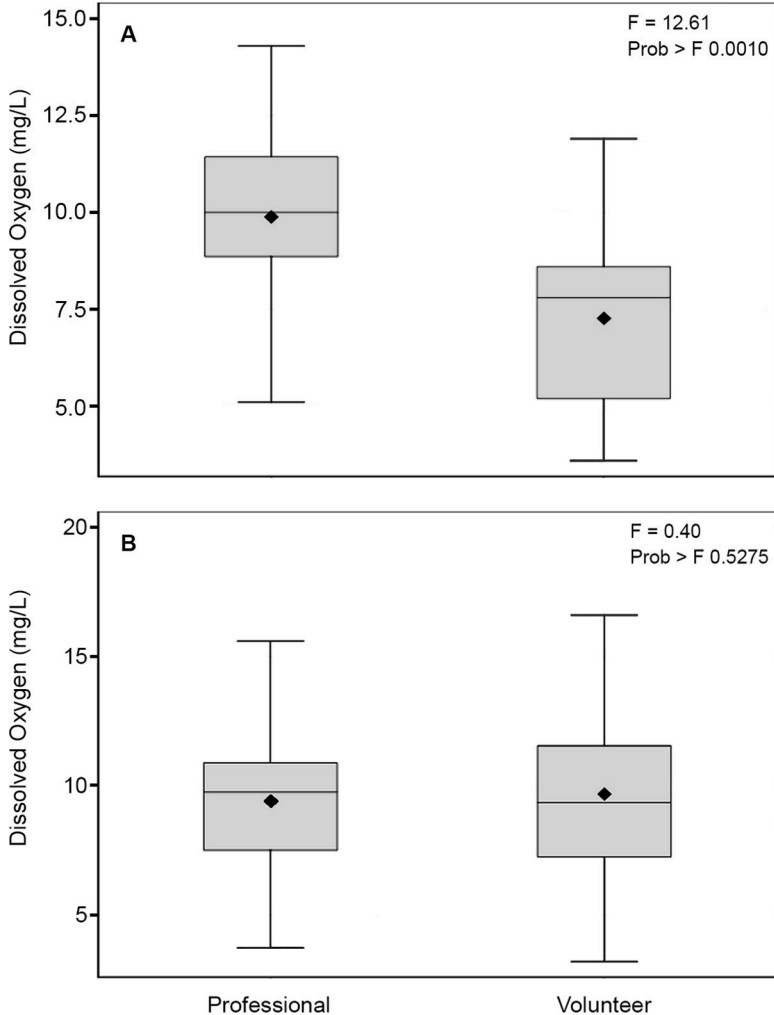

**Fig 2. DO bias adjustment, City of Denton dataset.** (A) Boxplot of all professional (COD) DO samples compared to all volunteer (TST) DO samples showing systematic bias across all years and stations. (B) All professional (COD) DO samples compared to all volunteer (TST) DO samples when systematic bias corrected by adding 2 mg/L to all volunteer samples.

the data distribution indicated, 2 mg/L was added to all volunteer DO samples which the bias was removed and there was no longer any significant difference between the volunteer and professional samples (Fig 2). For all results presented here for City of Denton data, the DO bias was removed with a +2 mg/L addition for all TST data. This was done as a form of data calibration, so that the results represent the actual variation in the dataset over time, station and source, rather than a systematic error between the source data groups [34].

## Results

### Statewide volunteer vs professional data

For the statewide analysis, a total of 234 TCEQ professional samples and 350 TST volunteer samples were selected for analysis. This resulted in 82 Station/Year ANOVAs, each one representing an analysis of variance between the volunteers and professionals, at one station for one year, for one of the parameters (DO, pH or conductivity) (Table 1). These final results included

**Table 1. Statewide ANOVA results by station and year.**

| Station | Year | DO | pH | Cond |
|---------|------|-----|-----|------|
| 15520 | 1999 | 0.947 | 0.236 | 0.069 |
| 15520 | 2000 | x | 0.342 | x |
| 15520 | 2001 | 0.944 | 0.748 | 0.411 |
| 15188 | 1996 | 0.447 | x | x |
| 11505 | 1992 | **0.001*** | x | x |
| 11505 | 1993 | **0.028*** | **0.009*** | x |
| 11505 | 1994 | 0.242 | 0.427 | x |
| 16404 | 1999 | **0.025*** | 0.063 | x |
| 13486 | 1992 | 0.696 | x | **< 0.0001*** |
| 13486 | 1993 | 0.662 | **0.0002*** | 0.162 |
| 12052 | 1998 | **0.008*** | **< 0.0001*** | 0.113 |
| 12052 | 1999 | 0.088 | x | x |
| 12052 | 2000 | 0.099 | x | x |
| 12052 | 2001 | 0.285 | **0.011*** | x |
| 17472 | 2012 | 0.690 | x | 0.848 |
| 17472 | 2013 | 0.115 | x | 0.391 |
| 17472 | 2014 | x | 0.389 | 0.845 |
| 17472 | 2015 | x | 0.520 | 0.652 |
| 17472 | 2016 | x | **0.038*** | 0.954 |
| 15964 | 2001 | 0.072 | x | 0.195 |
| 15964 | 2002 | 0.699 | x | 0.940 |
| 15964 | 2003 | x | x | 0.384 |
| 12500 | 1998 | **0.002*** | 0.145 | **< 0.0001*** |
| 12500 | 1999 | 0.290 | 0.647 | **< 0.0001*** |
| 12500 | 2001 | 0.0972 | 0.831 | 0.309 |
| 12500 | 2002 | 0.908 | 0.887 | 0.8801 |
| 12500 | 2003 | x | 0.743 | 0.065 |
| 12500 | 2004 | x | 0.746 | 0.341 |
| 12602 | 2012 | **0.022*** | x | 0.388 |
| 12602 | 2013 | x | x | x |
| 12602 | 2014 | **0.030** | 0.067 | 0.770 |
| 12602 | 2015 | 0.098 | x | 0.116 |
| 12602 | 2016 | 0.202 | 0.245 | **0.018*** |
| 17070 | 2003 | 0.747 | 0.217 | 0.360 |
| 12448 | 2005 | 0.625 | 0.068 | 0.271 |
| 12448 | 2006 | 0.892 | 0.201 | 0.986 |
| 12448 | 2007 | 0.960 | x | 0.525 |
| 12448 | 2008 | 0.059 | 0.758 | 0.363 |

Results of all ANOVAs ($Pr > F$) for each station and year by parameter. Values with an asterisk indicate a statistically significant difference between volunteer (TST) and professional (TCEQ) data. An "x" denotes a value removed due to missing, or non-normally distributed data. In cases where TST and TCEQ had different station IDs, the TCEQ station ID was used.

volunteer and professional data comparisons from 12 stations across 5 river basins, for a total of 38 station years of sampling data.

Our findings showed strong overall agreement, 81%, between volunteers and professionals for the entire statewide dataset for all three parameters. There was more agreement for conductivity

**Table 2. City of Denton ANOVA/KST results by station and year.**

| Station | Year | DO | pH | Cond |
|---------|------|------|------|------|
| 1 | 2009 | 0.4786 | 0.1389 | 0.1252 |
| 1 | 2010 | 0.9682 | 0.2898 | 0.6479 |
| 1 | 2011 | 0.6515 | 0.9251 | 0.9716 |
| 1 | 2014 | 0.6852 | 0.0815 | 0.8362 |
| 1 | 2016 | 0.9251 | 0.9639 | 0.3446 |
| 1 | 2017 | 0.5999 | **0.0366**\* | 0.2565 |
| 17 | 2017 | 0.4047 | 0.9639 | 0.9173 |
| 34 | 2009 | 0.9433 | 0.9639 | 0.5974 |
| 34 | 2010 | 0.1716 | 0.0815 | 0.5901 |
| 34 | 2014 | 0.9536 | 0.2700 | 0.2371 |
| 34 | 2016 | 0.7452 | 0.9639 | 0.3994 |
| 34 | 2017 | 0.2829 | 0.2700 | 0.6971 |
| 51 | 2009 | **0.0155**\* | 0.0815 | 0.9839 |
| 51 | 2010 | 0.5810 | 0.5176 | 0.2436 |
| 51 | 2011 | 0.4731 | 0.2700 | 0.8393 |
| 51 | 2014 | 0.7534 | 0.2898 | 0.3624 |
| 51 | 2017 | 0.1088 | 0.0996 | 0.7956 |
| 62 | 2009 | 0.1320 | 0.5176 | 0.8280 |
| 62 | 2010 | 0.6905 | **0.0366**\* | 0.8230 |
| 62 | 2011 | 0.9219 | 0.0649 | 0.6845 |
| 62 | 2016 | **0.0393**\* | 0.2700 | 0.3147 |
| 62 | 2014 | x | **0.0366**\* | x |
| 62 | 2017 | 0.0265 | **0.0366**\* | 0.9838 |
| 91 | 2017 | 0.1493 | 0.2106 | 0.5900 |

Results of all ANOVAs ($Pr > F$) for each station and year for DO and conductivity, and KST ($Pr>KSa$) results for all pH analyses. Values with an asterisk indicate a statistically significant difference between volunteer (TST) and professional (City of Denton) data. An "x" denotes a value removed due to data that was non-normally distributed.

at 86% (4 out of 28 ANOVAs significantly different), with pH at 79% (5 out of 24 ANOVAs significantly different) and DO the lowest at 77% agreement (7 out of 30 ANOVAs significantly different).

Of the 16 total ANOVAs that were significantly different, 11 of them occurred before the year 2000. From the time period 1992–1999 there were 26 ANOVAs run, meaning that 42% of them were significantly different. This is in contrast to the years 2000–2016, in which only five out of the 56 ANOVAs, 9%, run for that time period were significantly different. This indicates that relative accuracy between volunteers and professionals may have increased over time.

Although there were insufficient data to conduct statistical analyses, there also appeared to be differences among river basins in terms of percent agreement. The five Texas river basins for which there was at least one sampling station that fit the study criteria were the Sabine (100%; 8 out of 8 ANOVAs not significantly different), San Jacinto-Brazos (50%; 4 out of 8 ANOVAs), Brazos (57%; 8 out of 14 ANOVAs), Guadalupe (70%; 7 out of 10 ANOVAs) and Colorado (90%; 38 out of 42). Although the level of agreement for the Sabine river basin was especially high, there were only eight station/year analyses run for that basin, whereas the Colorado river basin had high overall agreement for the 42 station/year analyses run, more than the rest of the state combined, implying that the sampling data in that basin are highly consistent. These findings indicate that the location of the sampling station may impact the level of agreement between volunteer and professional datasets.

## City of Denton volunteer vs professional data

For local scale data collected within the City of Denton, there were a total of 159 volunteer and professional samples from 2009–2016, resulting in 70 analyses (ANOVAs for DO and conductivity and KST for pH), each one representing an analysis of variance between the volunteers and professionals at one station for one year and for one of the parameters (DO, pH or conductivity) (Table 2). Results included volunteer and professional data comparisons from 6 stations for a total of 24 separate station years of sampling data. Overall agreement was 91%. Agreement was 91% for DO with the bias adjustment, 83% for pH, and 100% for conductivity.

There was a higher percentage of agreement at local compared to statewide scales, likely due to decreasing temporal and spatial variation within and between the two sample datasets. Unlike at the statewide scale, the local analysis included only volunteer and professional samples that were paired within a five-day range of each other and at the exact same monitoring stations. By increasing analytical controls to include only volunteer and professional data that were more directly comparable than the statewide dataset, the relative accuracy of volunteer data increased as well.

Due to this direct comparability between volunteer and professional datasets on a local scale, a group analysis was conducted on the dataset as a whole to show variation between professionals and volunteers across all stations and years for each parameter. There was no significant difference between volunteer and professional data for conductivity ($N = 184$; $DF = 1$; Type III SS = 3.4060; Coeff. Var = 30.546; F Value = 0.40; Pr>F = 0.6230), or for DO ($N = 184$; $DF = 1$; Type III SS = 0.0330; Coeff. Var = 5.6719; F Value = 0.24; Pr>F = 0.5275) across all stations and years.

For pH, however, there was a significant difference (KS = 0.332; KSa = 4.508; D = 0.665; Pr>KSa < .0001). Although only 4 out of the 24 stations and years analyzed showed a significant difference (Table 2), the group analysis revealed significant variation between the volunteer and professional datasets as a whole. The empirical distribution of the pH data (Fig 3) indicates that this is likely caused by a lack of variability within the volunteer data as compared to the professional data. The volunteer data displayed a stairstep pattern with the majority of the pH levels concentrated in two discrete values rather than distributed continuously throughout the value range. The raw sampling data revealed that 90% of the volunteer pH samples were either a 7 or a 7.5, and all but six of the volunteer samples ended in either a zero or a five. This suggests that a lack of sensitivity in the volunteer equipment as compared to the professionals' equipment may have rendered the datasets less comparable.

## Experimental City of Denton volunteer vs professional data

For the City of Denton data collected as part of the experimental protocol, there were fewer samples (42 total samples; 21 each volunteer and professional) collected within a six-month period (Nov 2018 –May 2018) from 10 sampling stations across the City of Denton (4 of these stations new to this dataset). The smaller sample size allowed for a group ANOVA only, without individual station/year analyses.

As with the existing City of Denton data, there was no significant difference between volunteer and professional data for DO ($N = 42$; $DF = 1$; Type III SS = 3.9929; Coeff Var = 24.9227; F Value = 0.70; Pr>F = 0.4076) or conductivity ($N = 42$; $DF = 1$; Type III SS = 0.50317; Coeff Var = 8.5887; F Value = 1.60; Pr>F = 0.2132). Looking at the coefficient of variations for the existing data (DO = 30.546, Cond = 5.6719) compared to those of the experimental data there was slightly more variation for DO, and slightly less for conductivity. Also similar to the previous analyses, pH showed a significant difference between volunteer and professional data (KS = 0.286; KSa = 1.852; D = 0.571; Pr>KSa = 0.0021). The empirical distribution of the

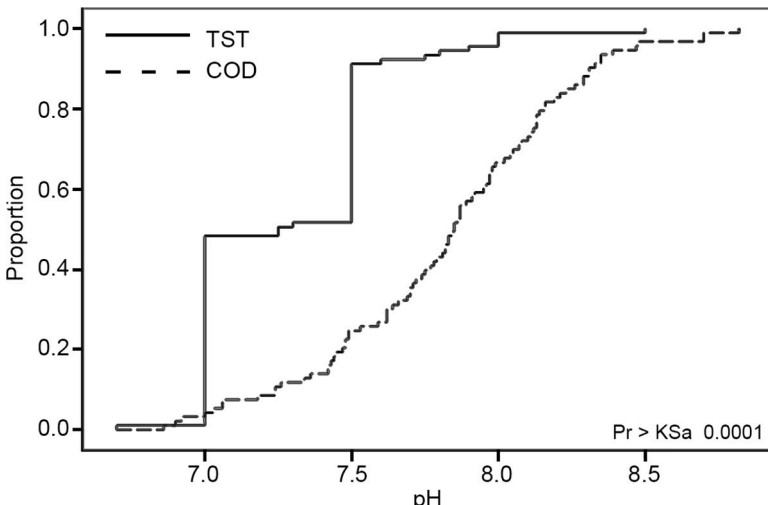

**Fig 3. Distribution of pH for existing City of Denton data.** The distribution of the volunteer (TST) and professional (COD) data at all stations for all years (2009–2017), with the KST statistic showing a significant difference between the two datasets.

experimental pH data (Fig 4) again revealed a clear lack of variability in the TST data as compared to the professional data. Visual assessment of the graph shows a slight increase in variability compared to the existing dataset, and examination of the raw sampling data support this with only 71% of the volunteer pH samples being either a 7 or a 7.5. Lack of sensitivity of the equipment is still evident as only two of these sample values end in a value other than zero or five. The increased variability between the existing data and the experimental data may reflect the addition of four new stations in the analysis. Overall, these results suggest no clear difference between the two datasets, demonstrating that existing data collected under routine citizen science program protocols can be comparable to data collected under an experimental research design.

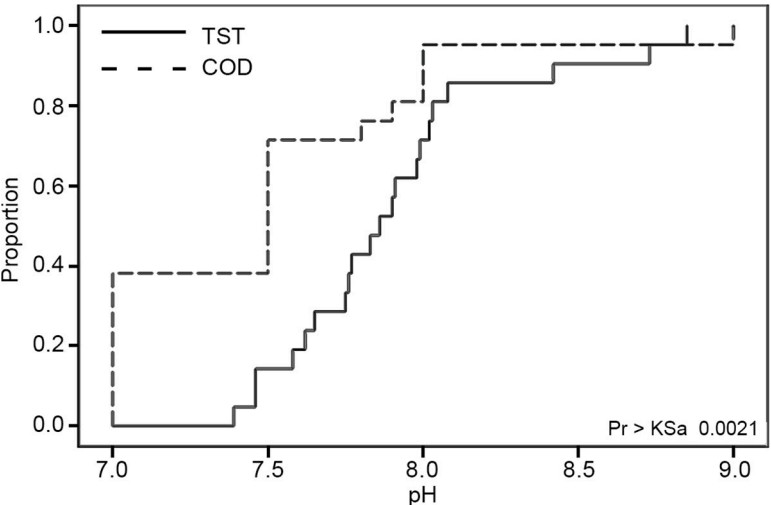

**Fig 4. Distribution of pH for experimental City of Denton data.** The distribution of the volunteer (TST) and professional (COD) data at all stations for 2017–2018, with the KST statistic showing a significant difference between the two datasets.

## Discussion

### Accuracy of volunteer data

Citizen science programs like TST have quality assurance protocols in place to maintain state and federal standards, and many have been collecting water monitoring data for decades. Despite these assurances, much of these data have been underutilized due to concerns about volunteer data accuracy [21, 26]. Specifically, high variability and the difficulty of working with unknown variables has led to concerns about consistent data quality [16, 35, 36]. Results from this study nonetheless show that long-running volunteer programs can maintain excellent agreement with professional data over time, indicating consistent high-quality data collection by certified citizen scientists.

It should be noted that in this and other comparison studies professional data are being held as the standard for accuracy, free of error and bias, which may not be the case [20, 4, 37]. Because professional data are the accepted standard for official use, studies in which the quality or accuracy of volunteer data are determined by comparison with data collected by professionals are an accepted method in the literature. Although most comparison studies determined that volunteer data showed good agreement with professional data, the majority did so through an experimental design with paired samples, and only on a smaller scale [33]. Findings from these controlled studies may not alleviate concerns about quality of existing data that volunteers collected throughout a program's entire history, and this may be especially true for large-scale datasets with more inherent variation. To the best of our knowledge this study is the first to utilize a volunteer program's long-term dataset to assess quality over multiple scales, and the findings provide unique insight into usability of citizen science data [33]. The results from this study demonstrate that monitoring data collected by citizen scientists can be comparable to data collected by professionals even with the natural variations that come with decades of monitoring on a statewide scale.

**Dedicated volunteer bias.**　The majority of the 2009–2016 City of Denton volunteer data were collected by a small number of dedicated TST volunteers that have been sampling as a group since the city began its partnership with TST. Four out of six stations analyzed and 63 out of the total 69 analyses were all sampled by this same group of volunteers, termed for this study as Group M, which has been sampling together consistently for 10+ years. As a result, their sampling data dominates the local volunteer dataset. The fact that Group M was the collector of most of the volunteer data from the City of Denton indicates that the results, for example the pH stairstep pattern and the 2 mg/L DO bias, could be due to the sampling equipment or methods used by all City of Denton volunteers, all TST volunteers, or it could be unique to Group M. A kind of dedicated-volunteer bias may be present here, which is likely true for many long-running programs. A similar bias may also be present in the statewide data as the analysis was limited to only those stations with 20 or more samples (or approximately 20 months of volunteer service). The only dataset that did not meet the minimum requirement of 20 samples were the experimental data, and only three of the 10 stations in this dataset were sampled by Group M. In this case, the coefficient of variation for the experimental pH data was higher than that of the existing data, which could be due to the experimental factor, or to the fact that they included more samples outside of Group M. This dataset was also the smallest, with sometimes only one paired sample coming from a station, so the results are not as likely to represent existing trends.

Ultimately, by selecting for long-term data this study could also be selecting for the more experienced volunteer groups, which could influence the accuracy of the data. However, this suggests that well trained volunteers with strict adherence to protocols and consistency lead to good and accurate data that is comparable to professional data, if not as high a resolution.

## DO bias and pH distribution

While we found excellent agreement at the local scale, group analyses revealed additional information. The first of these was a systematic bias in DO between volunteers and professionals of approximately 2 mg/L, across all stations and years, for both existing and experimental data. This is similar to results found by other researchers such as Dyer et al. [4] who found that volunteers consistently underestimated DO compared to professionals, and Safford and Peters [11] found that all three data sources (volunteer, professional, and USGS stations) reported DO values that were consistently lower than predicted values based on temperature-dependent equilibrium. As mentioned before however, professional data are not necessarily free from bias and could also be responsible for a portion of these differences.

The group analysis conducted on pH also revealed information that the station/year analyses did not. Although the majority of station/year analyses run for pH showed no significant difference, the group analyses showed that when combining all stations and years, the variation between volunteers and professionals was significant. The empirical distribution of the volunteer pH data revealed a stairstep pattern that was reflected in the raw data, with the volunteer pH data concentrated at whole number values rather than the continuous values seen in the professional data. This implies that the sampling methods or equipment used by the volunteers may not provide the same level of detail as that of the professionals. Since the majority of station/year analyses showed no significant difference between volunteers and professionals, however, it could mean that controlling for year and location could partly mitigate the differences in sampling protocol.

We were able to detect the DO bias and lack of variation in pH values because this portion of the analysis contained only volunteer and professional samples that were paired, meaning taken within a limited time frame at the same location. These findings suggest that paired samples are crucial for analyses when assessing volunteer relative accuracy as compared to professionals. Further studies utilizing paired sampling data from other volunteer and professional groups, which also includes details on specific sampling protocols used, are needed to determine possible causes for these differences in DO and pH values.

## Conclusions

By utilizing existing volunteer data to perform a post-hoc, multi-scaled analysis on a large-scale dataset, without experimental controls in place, this study was able to provide a reproducible framework for future analyses of other such existing datasets. Despite multiple sampling techniques conducted by numerous volunteer and professional entities over the years and a large geographic area with diverse ecosystems, the agreement between volunteers and professionals remained high. Our findings were also able to show that this strong agreement between volunteers and professionals holds true across multiple scales and increases as variation within the datasets is controlled for. Increased controls on multi-scale analyses identified a systematic bias in DO measurements and a pattern in pH data between volunteer and professional data which were consistent enough to be corrected for in further analysis. This consistent level of agreement between volunteers and professionals provides strong evidence that volunteer data can hold up to the most rigorous uses and suggests that a similar pattern could be found at other scales, possibly nationwide.

These can help inform pathways for volunteer data to be used alongside professional data in expanded capacities, areas where it may be needed most. Volunteers are often first responders after disasters or are granted access to areas that regulatory officials may not, like private lands or businesses, which means these volunteer-generated datasets may provide novel insights to researchers. Although citizen science has been an integral part of water quality

monitoring and management for decades, the long-term, quality-assured datasets these programs have generated may be unique resources that are, yet, still relatively untapped. A version of this analysis could be replicated on any existing volunteer monitoring dataset with a similar structure. Future research with existing, long-term volunteer data could further evaluate their uses when combined with professional resources in pollution remediation or emergency response efforts. Previous studies have indicated that volunteers can collect high quality data, and this study demonstrates that they have likely been doing so for many years.

## Supporting information

**S1 Data.**
(ZIP)

## Acknowledgments

The authors thank the Texas Stream Team staff and volunteers for their contributions to this research, along with the City of Denton for their assistance with the experimental portion of this study.

## Author Contributions

**Conceptualization:** Kelly Hibbeler Albus, Ruthanne Thompson.

**Formal analysis:** Kelly Hibbeler Albus, Forrest Mitchell, Alexandra G. Ponette-González.

**Investigation:** Kelly Hibbeler Albus.

**Methodology:** Kelly Hibbeler Albus, Ruthanne Thompson.

**Project administration:** Kelly Hibbeler Albus.

**Resources:** Ruthanne Thompson.

**Supervision:** Ruthanne Thompson.

**Validation:** Forrest Mitchell, Alexandra G. Ponette-González.

**Visualization:** James Kennedy, Alexandra G. Ponette-González.

**Writing – original draft:** Kelly Hibbeler Albus, Ruthanne Thompson.

**Writing – review & editing:** Kelly Hibbeler Albus, Ruthanne Thompson, Forrest Mitchell, James Kennedy, Alexandra G. Ponette-González.

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
