## [Decision Letter · Decision Letter 0]

18 Oct 2019

PONE-D-19-25900

Accuracy of long-term volunteer water monitoring data: A multiscale analysis from a statewide citizen science program

PLOS ONE

Dear Dr. Thompson,

Thank you for submitting your manuscript to PLOS ONE. After careful consideration, we feel that it has merit but does not fully meet PLOS ONE’s publication criteria as it currently stands. Therefore, we invite you to submit a revised version of the manuscript that addresses the points raised during the review process.

We would appreciate receiving your revised manuscript by October 31, 2019. To enhance the reproducibility of your results, we recommend that if applicable you deposit your laboratory protocols in protocols.io, where a protocol can be assigned its own identifier (DOI) such that it can be cited independently in the future. For instructions see: http://journals.plos.org/plosone/s/submission-guidelines#loc-laboratory-protocols

We look forward to receiving your revised manuscript.

Kind regards,

Md. Saifur Rahaman

Academic Editor

PLOS ONE

Journal Requirements:

2. In your Methods section, please include additional information about your dataset and ensure that you have included a statement specifying whether the collection method complied with the terms and conditions for the websites from which you have collected data.

3. We note that Figure [3] in your submission contains a  map image which may be copyrighted. All PLOS content is published under the Creative Commons Attribution License (CC BY 4.0), which means that the manuscript, images, and Supporting Information files will be freely available online, and any third party is permitted to access, download, copy, distribute, and use these materials in any way, even commercially, with proper attribution. For these reasons, we cannot publish previously copyrighted maps or satellite images created using proprietary data, such as Google software (Google Maps, Street View, and Earth). For more information, see our copyright guidelines: http://journals.plos.org/plosone/s/licenses-and-copyright.

You may seek permission from the original copyright holder of Figure(s) [3] to publish the content specifically under the CC BY 4.0 license. 

If you are unable to obtain permission from the original copyright holder to publish these figures under the CC BY 4.0 license or if the copyright holder’s requirements are incompatible with the CC BY 4.0 license, please either i) remove the figure or ii) supply a replacement figure that complies with the CC BY 4.0 license. Please check copyright information on all replacement figures and update the figure caption with source information. If applicable, please specify in the figure caption text when a figure is similar but not identical to the original image and is therefore for illustrative purposes only.

4. Please upload a copy of Supporting Information Figures and Tables which you refer to in your text on page 28.

Additional Editor Comments (if provided):

Dear Dr. Thompson,

Thank you for considering PLOS ONE for your manuscript submission. It has been forwarded to reviewers for their consideration, and the reviewers recommend reconsideration of your paper following major revision. I invite you to resubmit your manuscript after addressing all reviewer comments. When resubmitting your manuscript, please carefully consider all issues mentioned in the reviewers' comments, outline every change made point by point, and provide suitable rebuttals for any comments not addressed.

I look forward to receiving your revised manuscript as soon as possible.

Kind Regards,

Saifur Rahaman

Reviewers' comments:

Reviewer's Responses to Questions

**Comments to the Author**

1. Is the manuscript technically sound, and do the data support the conclusions?

Reviewer #1: Partly

Reviewer #2: Partly

Reviewer #3: Yes

2. Has the statistical analysis been performed appropriately and rigorously? 

Reviewer #1: Yes

Reviewer #2: Yes

Reviewer #3: Yes

3. Have the authors made all data underlying the findings in their manuscript fully available?

Reviewer #1: No

Reviewer #2: No

Reviewer #3: Yes

4. Is the manuscript presented in an intelligible fashion and written in standard English?

Reviewer #1: No

Reviewer #2: Yes

Reviewer #3: Yes

5. Review Comments to the Author

Reviewer #1: General Comments:

The study focuses on a very important evaluation of accuracy of the volunteer water quality monitoring program. It is an interesting study. The authors wanted to emphasize the importance of accuracy in the long term. However, water quality in natural water bodies vary widely even throughout the year. Long term accuracy doesn’t really guarantee short term suitability. During the time of pollution, it is not clear as to how the long term accuracy will ensure emergency response. The authors conducted a five day study to evaluate the short term implications. But, 21 samples on 10 different locations are good. However, two samples in each locations are not that high.

Specific Comments:

1. The quality of the figures are not good. Needs improvement.

2. Different headings used in different sections are confusing. For example “Volunteer and professional stream water quality monitoring” and “Statewide volunteer vs. professional data”. I couldn’t understand the differences between the two headings and the content inside them. Then you have another section with “City of Denton volunteer vs. professional data”. Please reconsider the headings. I think a generic name reflecting the big idea would be preferable.

3. In the section Statewide volunteer vs. professional data, statistical analysis is discussed. The authors mentioned about data collection and the years of data collection. However, the actual number of data used in the analysis is not clear. For example, 15 samples in 15 years are different to 15 samples in one year.

4. The data visible in the Table 2, are quite few. Analysis based on these data are difficult to conclude.

5. Experiments done by Group M were mentioned in the discussion. However, how many groups were there? There were not enough details shown.

6. Where there were sections identifying the bias in the water quality monitoring, the accuracy can be questioned. This should be mentioned in the conclusion as well.

Reviewer #2: The manuscript presents a comparative study between professional (state-wide) water monitoring results and the ones from volunteer groups. The paper first has focused on a state-scale and then narrowed down the comparison at the city scale. I have the following comments for the authors to consider:

•The reviewer is still unsure about the main take away from this comparative study. Do authors want to advocate on sampling being done by volunteer group? And if so, upgrading their sampling methods and equipment? Is this advocated as part of a larger scale transition to a “social innovation” paradigm with reduced role and responsibility for the Governments? In that case who can be held accountable in case of monitoring errors and health-related consequences? The reviewer was in search of a wider motivation for this study.

•What is the importance of looking into these differences between the two groups of samples in “long-term”? Was the main purpose to create a larger data set? I believe that the authors could have tried to divide data to a number clusters based on sampling methods equipment characteristics and then to only compare the two groups cluster-by-cluster.

•I suggest that the authors add a table summarizing the sampling characteristics in each monitoring group over the years, and if there has been any update during this time on sampling methods and equipment.

•It was not clear why the focus of the comparison was on DO, pH, and conductivity? What about BOD, Nitrate, phosphorous, and bacteria?

•The removal of bias/error from DO samples from Volunteers: How did the authors reach to a conclusion to add 2 mg/L to all samples? How did the authors arrive at this number?

•Despite an emphasis expressed at the beginning of the article, not much of analysis was conducted on “conductivity” criteria.

•Minor comments:

- There were no page numbers.

- Figure sizes shall be reduced.

Reviewer #3: This study assesses the reliability of large-scale volunteer water quality data by comparing it corresponding professionally collected data set. This is a well-written manuscript with systematic presentation and interpretation of data that can be followed quite well. I have a couple minor comments. DO bias adjustment was not highlighted in the Abstract. Please elaborate a little more (paragraph after Table 2) “By increasing analytical controls to …….. relative accuracy of volunteer data increased”.

6. PLOS authors have the option to publish the peer review history of their article (what does this mean?). If published, this will include your full peer review and any attached files.

Reviewer #1: No

Reviewer #2: No

Reviewer #3: No

---

## [Author Response · Author response to Decision Letter 0]

18 Nov 2019

We have uploaded a document entitled - Response to Reviewers. We are hopeful that document meets this requirement?!

---

## [Editor Report · Decision Letter 1]

23 Dec 2019

Accuracy of long-term volunteer water monitoring data: A multiscale analysis from a statewide citizen science program

PONE-D-19-25900R1

Dear Dr.Thompson,

We are pleased to inform you that your manuscript has been judged scientifically suitable for publication and will be formally accepted for publication once it complies with all outstanding technical requirements.

With kind regards,

Md. Saifur Rahaman

Academic Editor

PLOS ONE
---

## [Editor Report · Acceptance letter]

3 Jan 2020

PONE-D-19-25900R1 

Accuracy of long-term volunteer water monitoring data: A multiscale analysis from a statewide citizen science program 

Dear Dr. Thompson:

I am pleased to inform you that your manuscript has been deemed suitable for publication in PLOS ONE. Congratulations! Your manuscript is now with our production department. 

With kind regards,

on behalf of

Dr. Md. Saifur Rahaman 

Academic Editor

PLOS ONE